# Effectiveness and Safety Profiles of Biological Therapies in Inflammatory Bowel Disease: Real Life Data from an Active Pharmacovigilance Project

**DOI:** 10.3390/biomedicines10123280

**Published:** 2022-12-18

**Authors:** Maria Antonietta Barbieri, Anna Viola, Giuseppe Cicala, Edoardo Spina, Walter Fries

**Affiliations:** Department of Clinical and Experimental Medicine, University of Messina, 98125 Messina, Italy

**Keywords:** biologics, inflammatory bowel disease, pharmacovigilance, treatment persistence, adverse drug reaction

## Abstract

Post-marketing surveillance is essential to evaluate the risk/benefit profile of drugs; however, pharmacovigilance studies comparing persistence and safety of biologic therapies in patients with inflammatory bowel disease (IBD) are scant. The aim of this study was to prospectively investigate persistence together with safety profiles of biologics in a cohort of patients diagnosed with Crohn’s Disease (CD) or ulcerative colitis (UC) followed by the IBD unit of Messina and treated with infliximab (IFX), adalimumab (ADA), golimumab (GOL), vedolizumab (VED), and ustekinumab (UST) from 2017 through 2021. Descriptive and treatment persistence analyses with predictors for discontinuation and occurrence of adverse drug reactions (ADRs) were performed. A total of 675 IBD patients were enrolled. A higher persistence rate was noted for UST and ADA in the first year (83.8% and 83.1%, respectively) and for IFX in the fifth year of treatment (58.1%). GOL, VED, and UST—all used as second/third-line therapies—seemed to have a higher risk of non-persistence than IFX (in order HR: 2.19; CI 95%: 1.33–3.61, 1.45; 1.04–2.04, 2.25; 1.25–4.07) as well as switchers and those who had at least one ADR (18.1; 13.22–24.68 and 1.55; 1.20–1.99, respectively). The reported ADRs, which were generally mild–moderate, were largely known. However, real-world data should be implemented to further study undetected safety concerns, including risk of malignancy.

## 1. Introduction

Inflammatory bowel diseases (IBD), i.e., Crohn’s disease (CD) and ulcerative colitis (UC), are characterized by chronic inflammation of the gastrointestinal (GI) tract with an immune-mediated pathogenesis [1]. The clinical course of IBD with periods of relapse and remission leads to several complications, including an impaired quality of life, and the need for hospitalization and surgery [2]. Currently, IBD affect a growing number of people worldwide, with more than 6.8 million cases, and the number will increase exponentially over the next decade [3]. New data suggest that around 0.2% of the European population suffers from IBD at the present time [4] and approximately 260,000 subjects are affected by IBD in Italy (0.4%) [5].

Concerning management of IBD, topical and systemic treatments, such as aminosalycilates, corticosteroids (CSs), antibiotics, immunomodulators including thiopurines and methotrexate, and cyclosporine, have been used for many years [6,7]. Since 1995, new therapies have been developed with the approval of the monoclonal antibody against tumor necrosis factor-α (anti-TNFα), infliximab (IFX), leading to a new era for the treatment of IBD [8]. The following biologic agents are currently available in Italy: anti-TNFα agents (IFX and adalimumab (ADA)) approved for CD and UC, and golimumab (GOL) approved only for UC; the anti-integrin drug vedolizumab (VED); and the anti-interleukin (IL) 12/23 molecule ustekinumab (UST) [9]. Biological therapies are now the mainstay in the treatment of IBD, reducing hospitalizations and surgeries and inducing higher rates of remission and response than previous treatments with mucosal and histological healing [10,11]; however, many patients do not respond to induction doses or lose response after an initial improvement [12,13]. The introduction of different treatment options makes it difficult to manage IBD patients, especially for the choice of the best biological therapy. In this context, real-world comparative data that analyzes the effectiveness of biologics could lead the physician to choose the most effective therapy in patients affected by IBD [14]. Nonetheless, the increased use of these drugs has been associated with the onset of some emerging safety issues, such as hypersensitivity and immunogenicity reactions, infections, vasculitis, and some types of malignancies, for which an adequate selection of patients eligible for treatment and post-marketing monitoring, especially in long-term use, is required [15,16,17]. In this scenario, active pharmacovigilance programs become crucial for improving the detection of adverse drug reactions (ADRs), including serious ADRs (SADRs) [18,19]. For all the above reasons, the aim of this study was to evaluate the effectiveness and safety profile of biologics for the treatment of patients affected by IBD during a prospective pharmacovigilance study.

## 2. Materials and Methods

### 2.1. Study Design and Data Collection

A prospective observational study was carried out from January 2017 to December 2021 at the IBD unit of University Hospital of Messina on the basis of an active pharmacovigilance project created to improve knowledge on the effectiveness and safety of biologic agents approved for inflammatory chronic conditions, including IBD. The study protocol was approved by the local Ethics Committee of Messina University Hospital (protocol number 125/2016).

All patients ≥ 16 years with a confirmed histological diagnosis of CD or UC followed by the IBD unit of Messina, having been treated with at least one biologic drug (IFX, ADA, GOL, VED, and UST) at the start of the study or having commenced a biologic during the study period, were enrolled, monitored, and evaluated in the study. The date of the first biologic drug prescription during the study period was considered the “index date” for each patient. Demographic, clinical, and disease-related data were collected in a dedicated database. A monitor, a specialist in clinical pharmacology who had received specific training in pharmacovigilance, was assigned to help the gastroenterologist physician in the creation of the database in accordance with Good Clinical Practice recommendations while a patient encrypted code was used to maintain the anonymity of patients in agreement with the Declaration of Helsinki. Specifically, the following information was collected: age, sex, body mass index (BMI), clinical diagnosis based on the Montreal classification [20], disease duration, smoking habits, comorbidities, extraintestinal manifestations (EIMs) coded by the International Statistical Classification of Diseases and Related Health Problems 10th Revision (ICD-10), drugs used, including all drugs taken concomitantly by each patient, switch/swap to another biologic drug, potential primary or secondary failures, and the possible onset of ADRs. 

Patients were defined as stoppers if they discontinued treatment, including when biologics had not been taken within the recommended time. Additionally, patients were considered switchers if they started another biological drug during the study follow-up period compared to the one used at the index date. The clinical pharmacologist supported the physician in identifying failures and potential ADRs through an accurate analysis of medical records. Patients underwent scheduled follow-up visits with the evaluation of clinical scores to establish the state of activity of IBD, disease response, or potential remission. Primary failure was defined as a lack of therapeutic drug response by the induction phase, whereas secondary failure was defined as a loss of clinical response after an initial improvement [21]. All ADRs were collected in accordance with the definition reported in the Guideline on Good Pharmacovigilance Practices as “a response to a medicinal product which is noxious and unintended”, which means that a causal relationship between a medicinal product and an adverse event is at least a reasonable possibility [22]. For each observed ADR, the investigator, both physician and clinical pharmacologist, filled in the suspected ADR reporting form of the Italian Medicines Agency with a detailed description that included time to onset and recovery, seriousness, and outcome. The identification of SADRs was collected in accordance with the Guideline on Good Pharmacovigilance Practices and it was related to life-threatening or fatal conditions, hospitalization, persistent or significant disability, or a congenital anomaly/birth defect, surgery, and death, or to other important medical events including malignancies [22].

### 2.2. Data Analyses

Descriptive statistical analyses were performed to evaluate clinical and demographic characteristics of enrolled patients at the index date by diagnosis and by each biologic of interest. Each patient was assigned to a specific treatment. Medians with interquartile ranges (Q1–Q3) for continuous variables and absolute and percentage frequencies for categorical variables were estimated. Since a non-normal distribution of some of the numerical variables was verified using the Kolmogorov–Smirnov test for normality, a non-parametric approach was adopted. The Mann–Whitney U test for independent samples and two-tailed Pearson’s chi-squared test were performed to compare continuous and categorical variables, respectively; moreover, an analysis of switch/swap in term of frequency was conducted between various biological drugs to quantify the switching pattern during the entirety of follow-up. Thereafter, the total years of treatment and an analysis of all primary/secondary failures and all ADRs expressed as number of failures and ADRs adjusted to 10 years of treatment was carried out by taking into account the total years of treatment for each biologic, including all patients treated at least once during the follow-up period to detect the rate of failures and ADRs for each drug. Furthermore, a sub-analysis of ADRs by seriousness and by type was performed to evaluate the higher frequency of some ADRs for each biologic of interest.

For each biologic user group, the crude risk of stopping was evaluated by Kaplan–Meier analyses and log-rank tests. Multivariate Cox proportional hazards models were carried out in order to assess factors associated with the non-persistence of biologics. Baseline covariates including gender, age, the period from diagnosis date to first biologic use (<1 year vs. ≥1 year), naïve or not to biological treatment, and number of previous biological therapies were used in the adjusted analysis as time-fixed covariates. The results were shown as hazard ratios (HRs) with corresponding 95% confidence intervals (CIs). *p* values < 0.05 were considered statistically significant. All statistical analyses were conducted using SPSS version 23.0 (IBM Corp. SPSS Statistics, Armonk, NY, USA).

## 3. Results

### 3.1. Study Population

From 2017 to 2021, a total of 675 IBD patients were enrolled, including 392 (58.1%) with CD and 283 (41.9%) with UC. Patients were mostly males (*n* = 395; 58.5%) with a median age (Q1-Q3) of 44 (29–58) years. Nevertheless, UC patients were older than CD patients (*p* = 0.009). At the index date, there were 429 (63.6%) naïve patients mainly affected by CD (*p* = 0.013). Patients with CD were more frequently smokers (*p* < 0.001). Furthermore, comorbidities such as diabetes, hypertension, and cardiovascular and chronic kidney disease were mainly present in UC patients (*p* < 0.05 for all comparisons). In general, EIMs were more frequent in CD than in UC patients (*p* = 0.001). Mesalazine was mostly used in association with biologics in UC patients (*p* < 0.001) (Table 1).

Considering biologic users, ADA and IFX were the most prescribed drugs (*n* = 269; 39.9% and *n* = 225; 33.3%, respectively), followed by VED (*n* = 145; 21.5%); in detail, IFX and VED were mainly used in UC patients (*n* = 152; 67.6% and *n* = 93; 64.1%, respectively), while ADA prevailed in CD subjects (*n* = 249; 92.6%). Furthermore, 92.9% and 78.1% of patients in treatment with IFX and ADA were naïve, respectively (Table 2).

A total of 223 patients (33.0%) switched/swapped to another biologic, of which 153 switch/swapped due to therapeutic failure (68.6%) and 66 for ADR (29.6%). Specifically, 59 patients (26.5%) were swappers from ADA to UST, while 44 patients (19.7%) swapped from IFX to VED (Table 3).

During the study period, the total years of treatment were 890 years for ADA, 710.8 years for IFX, 336 years for VED, 90.3 years for UST, and 61.3 years for GOL. A higher incidence of failures calculated on 10 treatment-years was shown for GOL (3.6/10 treatment-years), while a higher rate of ADRs was noted for UST (2.5/10 treatment-years) (Table 4).

Considering the first-line therapies, a higher incidence of ADRs and failures was reported for VED (both with 6.3/10 treatment-years); IFX and UST had a higher rate of ADRs when they were used as second- (both with 1.7/10 treatment-years) and third-line therapies (IFX, 2.9/10 treatment-years; UST, 3.7/10 treatment-years), while GOL had a higher incidence of failures, especially as a third-line treatment (11/10 treatment-years) (Appendix A).

### 3.2. Persistence of Treatment

The Kaplan–Meier analysis showed that after 5 years about half of the patients were persistent on their first-line treatment. Patients treated with IFX and ADA commonly continued their initial therapy longer compared to other drugs. Considering the first year of treatment, a higher persistence rate was noted for UST and ADA (83.8% and 83.1%, respectively); thereafter, ADA continued to have the highest persistence rate throughout year 4, but patients most likely to persist with treatment in the fifth year were IFX users (58.1%) (Figure 1).

When separately analyzing persistence of first-line and second-/third-line treatments (Appendix A), no difference was noted between the various treatment options—neither between first-line agents i.e., IFX and ADA, nor between second-line treatments; numerically, second-line treatments showed shorter persistence periods than first-line treatments. Finally, after analyzing persistence by disease, no difference was found in CD considering every kind of biologic used, whereas in UC a significant (*p* = 0.028) difference was noted with the highest persistence rates for IFX followed by VED and ADA, and the lowest persistence rates for GOL; in UC, UST was not considered due to a lack of indication for UC in the study period (Appendix A).

After adjusting for predictors by applying a Cox regression analysis, GOL, VED, and UST users seemed to have a higher risk of non-persistence than IFX users (in order HR: 2.19; CI 95%: 1.33–3.61, 1.45; 1.04–2.04, and 2.25; 1.25–4.07), while ADA showed a comparable risk; moreover, females had an increased risk of non-persistence compared to males (1.28; 1.02–1.61), while patients aged 31–65 had a reduced risk of non-persistence compared to younger patients (0.68; 0.51–0.92). Subjects affected by CD and by EIMs had a lower risk of non-persistence (0.66; 0.48–0.90 and 0.66; 0.49–0.89, respectively), such as naïve patients and CS users (0.70; 0.54–0.89 and 0.55; 0.30–0.99, respectively). Conversely, switchers and those who had had at least one ADR were significantly associated with increased risk of non-persistence (18.1; 13.22–24.68 and 1.55; 1.20–1.99, respectively) (Figure 2).

### 3.3. Description of Adverse Drug Reactions

A total of 249 ADRs were reported during the study period. There were 71 (28.5%) SADRs mainly in patients treated with UST (1.0/10 treatment years), GOL (0.7/10 treatment-years), and VED (0.6/10 treatment-years). No difference was shown in the onset of SADRs between patients with EIMs (*n* = 9; 7.6%) and without EIMs (*n* = 62; 10.5%, *p* = 0.437). There were 24 patients aged 18 to 30 years with at least one SADR (9.4%), there were 38 patients aged 31 to 65 years with at least one SADR (6.7%), and there were 11 elderly patients with SADR (8.4%). No statistically significant differences were shown in the occurrence of SADRs by age group (*p* = 0.390). The most observed ADRs were: infections (*n* = 75; 30.1%), including herpes zoster, COVID-19, and dental abscess, which occurred mainly in UST-treated patients (1.1/10 treatment-years), especially as third-line therapy; infusion-related reactions with dyspnea, chest pain, hyperhidrosis, and tachycardia (*n* = 42; 16.9%), and joint disease mostly related to IFX infusion (0.5/10 treatment-years, *p* < 0.001 and 0.1/10 treatment-years; *p* = 0.025, respectively); and skin reactions (*n* = 38; 15.3%) such as erythema, pruritus, and psoriasis, which were commonly associated with ADA (0.3/10 treatment-years, *p* < 0.001) when used as a first-line treatment. The onset of malignancies was reported in 14 patients (5.6%), especially colorectal cancer (*n* = 4) followed by skin and liver cancer (*n* = 2, both), with a higher frequency in VED-treated patients (0.3/10 treatment-years; *p* = 0.002)—particularly when used as a third-line treatment (Table 5).

A detailed description of ADRs based on the diverse line therapies is shown in Appendix A. During follow-up, two of the SADRs had a fatal outcome. The first case involved a 53-year-old male patient affected by CD for 36 years that had been treated with VED and mesalazine from May 2017 who presented at the emergency department for abdominal pain in February 2018 due to metastatic colon cancer confirmed by histological examination. The last infusion of VED was administered in November 2017 and was not resumed, and the patient died in August 2018. The second case occurred in a 66-year-old male patient with a diagnosis of UC for 1 year under treatment with IFX biosimilar and mesalazine since March 2018. The last infusion of IFX was carried out in June 2018; in July 2018, the patient died from massive pulmonary embolism but no more in-depth information about the definitive cause was available; however, the Naranjo Algorithm showed a possible association between ADR and drug administration for both fatal cases.

## 4. Discussion

This is, to the best of our knowledge, the first prospective population-based study aimed at investigating persistence together with safety profiles of all biologics approved for IBD in Italy. Former reports on persistence were limited to time frames before the introduction of VED or UST or very shortly thereafter [23,24,25,26,27], thus including mainly anti-TNF agents. 

In the present observational study, more patients diagnosed with CD than those with UC were subject to biologic therapy, thereby indicating a more frequent need for such therapies in the former in accordance with the literature [25]. Concerning biologic use at the index date, IFX and ADA were the most prescribed first- and second-line treatment drugs due to the availability of less costly biosimilars, followed by VED and UST. Overall, 81% of our patients were persistent with the chosen therapy at the end of the first year; hence, the non-persistence rate of 19% was in line with the rates shown in previous studies [23,24,26], but lower than it was observed in two studies reporting a non-persistence less than 50% at and after 1 year [25,28]. The results for ADA and IFX were very similar to a Japanese and a French study [27]. Our data for VED as second-/third-line treatment showed better results after 1 year of therapy compared with a small study from Germany [29]. Importantly, data related to the most recently approved biologic for CD, UST, showed a similar non-persistence rate of 14.9% in the first year of treatment, which is in line with two former reports [30,31]. Considering persistence rates over a 5-year treatment period in our patients, IFX and ADA persistence dropped by 19% and 29%, respectively, when used as first-line treatments, and they dropped by 44% and 33%, respectively, when used as second-/third-line therapies. By separately analyzing 5-year persistence for disease type in CD, the reduction of persistence to IFX and ADA was 18.5% and 28.4%, respectively, which is quite similar to a Korean study [24]. In UC, the 5-year reduction of persistence was 25.5% for IFX, whereas for ADA the 5-year data were not available as reported in the aforementioned Korean paper [24]. In a former retrospective US study carried out between 2008–2015, even higher 5-year reduction rates of persistence were reported for CD and UC [25]. Substantially, all these data are consistent with the progressive loss of response of monoclonal-based biologic therapies. Slight differences in favor of IFX or ADA in terms of persistence may depend on the type of monitoring of patients, but apparently seems to not depend on immunogenicity as expected.

A Cox regression analysis showed that patients treated with GOL, VED, and UST seemed to have a higher risk of non-persistence than IFX users, while for ADA the risk was similar, as shown in a previous study [25]. Again, IFX and ADA are most frequently used as first-line therapies due to the availability of less costly biosimilars, whereas GOL, VED, and UST—not only because of their high costs—are considered second- and third-line therapies, and we know from many randomized clinical trials (RCTs) that naïve patients generally exhibit a better response than biologic-experienced patients [32,33,34,35]. In a recent paper [36], UST (with data from 2 years available) performed best in luminal CD, whereas VED was superior in UC. Interestingly, again ADA in UC did not achieve the 5-year observation limit. Naïve patients had a reduced risk of non-persistence, although a similar trend in persistence between bio-naïve and bio-experienced patients with UC was shown in the literature [37]; moreover, female patients seem to be at more risk of non-persistence than males possibly due to the higher occurrence of ADRs due to physiological differences [23,38,39,40]. Another relevant risk was shown for younger patients probably due to a possible pregnancy decision or lifestyle changes, including moving to other cities for new job opportunities [25,41]. Furthermore, switchers/swappers and those who experienced ADRs were associated with a higher risk of non-persistence: 33% of patients switched to another biologic at least once, especially patients treated with anti-TNFα, as observed in other studies [28,42,43]. The treatment failures were mainly reported for GOL, which consistently remained the least frequently used anti-TNFα drug in UC and was used more frequently as a second- and third-line therapy [44]. Recent studies confirm that UC patients treated with GOL after previous failure of other anti-TNFα drugs had significantly worse outcomes [45,46], whereas the use of the i.v. anti-TNF IFX after failure of s.c. GOL or ADA led to better results [47].

Considering ADRs in the year before the index date of the present study, only 7 ADRs were spontaneously reported for biologics in the same center: most were SADRs (*n* = 6; 85.7%) and principally related to IFX (*n* = 5; 71.4%). In the present study, a total of 249 ADRs were reported during the study period, thereby underlining the importance of active pharmacovigilance projects. Our findings are consistent with the Summaries of Product Characteristics (SmPCs) available at the time of the study. Concerning SADRs, although not significant, there was a numerical increase with second-/third-line therapies. SADRs include hospitalizations and surgeries and difficult-to-treat patients, which does not necessarily imply a drawback for certain biologics.

The most common ADRs were infections, especially with third-line treatments—except for VED—but no significant risk was found for any agent. Controversial data on infections have been reported in the literature: biologics that suppress the immune system have been associated with increased rates of infections in several studies [48,49,50], but a meta-analysis on anti-TNFα drugs in patients with CD demonstrated a decreased incidence of SADRs, including infections [51]. The more recently available biologics, such as VED and UST, did not seem to show an increased risk of serious infections compared to anti-TNFα agents [52,53]. Additionally, IBD patients had several intrinsic risk factors that play a key role in the development of infections, including malnutrition, surgery, and “leaky gut syndrome”, with consequent increased pathogen exposure and bacterial translocation, especially when co-treated with steroids that could contribute to the observed increase in infection rates [54,55,56]. Anti-TNFα agents, particularly ADA, followed by UST and VED, play a role in the onset of dermatological pathologies, including urticaria, erythema, and dermatitis [57]; moreover, our data showed a higher incidence of malignancies with VED when adjusted on 10 years of treatment. Chronic inflammation is more frequently related to colorectal cancer development and extraintestinal neoplasms in IBD patients than in the general population [58,59]. Nevertheless, immunosuppressive treatment, including biologics, exhibit carcinogenic properties [60]. A prospective study showed that extensive UC and perforating CD significantly increased the risk for any cancer in IBD; moreover, the severity of IBD was reported as a risk factor for extracolonic cancer, while the IBD duration significantly increased the risk of cancers involving the urinary tract and the skin in IBD, especially in UC [59]. The cancer risk in IBD should be carefully assessed and monitored in each patient at each time of the IBD course, thereby considering multiple variables, such as intrinsic IBD risk factors (e.g., CD phenotype, UC extent, and severity), but also immunomodulator use [61,62]. However, no higher risk of malignancies or their recurrence was shown for VED compared with anti-TNFα in the literature [63,64], although the follow-up period is likely too short to confirm or confute the malignancy risk. In the present paper, the increased risk for malignancies of VED may reflect a more difficult-to-treat patient who failed to respond to prior anti-TNFs and thus has a more prolonged history of active disease.

### Strengths and Limitations

Our study has strengths and limitations. Among the former, there is the prospective nature of data collection together with the active pharmacovigilance approach in a real-world setting, including almost all available biologic agents. Passive pharmacovigilance captures mostly SADRs while neglecting ADRs, which are considered to be of minor importance. The management of IBD has been greatly revolutionized by the availability of biologics. Although both the efficacy and safety of these drugs have been established in many clinical trials [65,66,67,68], the benefit/risk profile remains controversial due to the different persistence rates based on the risk of immunogenicity and SADRs, such as the onset of infections and neoplasms. Consequently, the key strength of this study is that real-world analyses increase awareness on biologic use in IBD; moreover, active pharmacovigilance plays a crucial role in detecting ADRs and SADRs [69,70] and post-marketing activities are essential to implement the safety profile of these new treatments, thereby reducing any under-reporting phenomena and resulting in therapy optimization in clinical practice [18,71]. The choice of a specific biologic on the index date may be related to the patient’s medical history, but, in effect, depends on regional directives that regulate biologic and biosimilar prescriptions. However, clinical characteristics associated with the geographical area should not modify the estimated effectiveness and safety profile of treatments. Furthermore, the Cox regression analysis took several predictors into account, including disease duration, first biologic use, and concomitant therapies, which can influence the persistence rate and the ADR occurrence while avoiding any confounding factors. 

Nevertheless, this study has some limitations. First, the non-persistence rate may have increased over time due to the start of treatment with small molecules, such as tofacitinib, which has been available for the past two years for UC. Additionally, patients lost to follow-up could create a bias in persistence rates due to discontinuation of biologics for prolonged remission or transfer to other gastroenterological units. Furthermore, it was not easy to compare the prescription patterns of biological agents because of the recent approval of GOL, VED, and UST for CD and UC patients; in particular, the long-term persistence and switching rate of UST could not be assessed. Moreover, switch/swap from second- to third-line treatment was not included (Table 3); however, the number of patients who switched twice or more was so small that this hardly influenced our results.

## 5. Conclusions

Considering the increasing use of biologics in IBD, pharmacovigilance activities have an important role in significantly improving awareness of the persistence and safety profiles of these drugs in clinical practice. Overall, biologic drug persistence profiles suggested a high proportion of patients continued on their initial biologics for 1 year and approximately half of patients were persistent after 5 years of treatment. Patients treated with GOL, VED, and UST seemed to have a higher risk of non-persistence than IFX users. The reasons for non-persistence are likely to be age and gender, but also those who switch/swap and those who experienced ADRs. The reported ADRs, which are generally mild-to-moderate and mostly related to infections and skin disorders, were largely known, but real-world data should be implemented in investigating undetected safety concerns, including risk of malignancy.

## Figures and Tables

**Figure 1 biomedicines-10-03280-f001:**
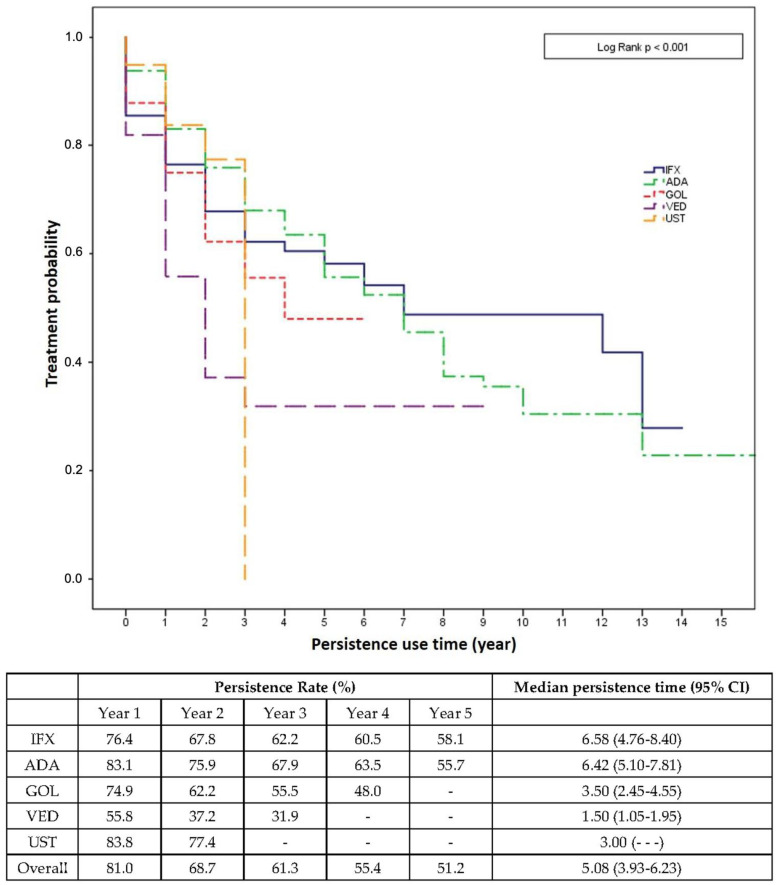
Kaplan–Meier curve for the persistence of biologics in patients with IBD. Abbreviations: ADA = adalimumab; CI = confidence interval; GOL = golimumab; IFX = infliximab; VED = vedolizumab; UST = ustekinumab.

**Figure 2 biomedicines-10-03280-f002:**
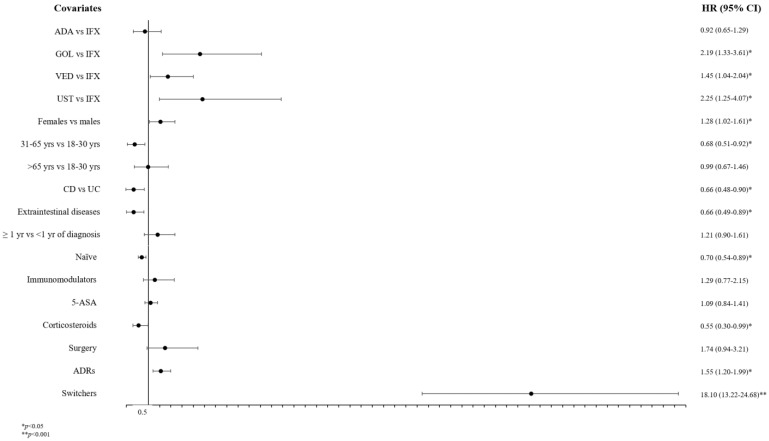
Predictors of non-persistence of biologics in patients with IBD. Abbreviations: 5-ASA = mesalazine; ADA = adalimumab; ADRs = adverse drug reactions; CD = Crohn’s disease; GOL = golimumab; IFX = infliximab; VED = vedolizumab; UC = ulcerative colitis; UST = ustekinumab; yrs = years.

**Table 1 biomedicines-10-03280-t001:** Basal characteristics of patients affected by IBD divided by diagnosis.

Characteristic	CD (*n* = 392)	UC (*n* = 283)	*p* Value *	Total (*n* = 675)
Sex, *n* (%)				
Females	158 (40.3)	122 (43.1)	0.466	280 (41.5)
Males	234 (59.7)	161 (56.9)		395 (58.5)
Median age (Q1–Q3)	42.5 (28–57)	45 (31–60)	**0.009**	44 (29–58)
Median age at diagnosis (Q1–Q3)	28 (20–41.8)	34 (24–47)	**<0.001**	31 (21–44)
Naïve, *n* (%)	265 (67.6)	164 (58.0)	**0.013**	429 (63.6)
Disease duration, median (Q1–Q3)	8 (3–16)	7 (2–14)	0.184	8 (3–15)
Surgery, *n* (%)	142 (36.2)	12 (4.2)	**<0.001**	154 (22.8)
Smoking, *n* (%)				
Smoker	121 (30.9)	24 (8.5)	**<0.001**	145 (21.5)
Ex-smoker	48 (12.2)	58 (20.5)		106 (15.7)
Non-smoker	223 (56.9)	201 (71.0)		424 (62.8)
BMI, median (Q1–Q3)	23 (20.9–26.3)	24.1 (21.7–27.1)	0.131	24 (21.3–26.4)
Comorbidities, median (Q1–Q3)	0 (0–1)	1 (0–2)	0.490	1 (0–2)
Comorbidities, *n* (%)				
Anxiety–depressive disorders	20 (5.1)	8 (2.8)	0.144	28 (4.1)
Cardiovascular disease	25 (6.4)	31 (11.0)	**0.033**	56 (8.3)
Chronic kidney disease	3 (0.8)	11 (3.9)	**0.005**	14 (2.1)
Chronic pulmonary disease	10 (2.6)	5 (1.8)	0.495	15 (2.2)
Diabetes	10 (2.6)	22 (7.8)	**0.002**	32 (4.7)
Dyslipidemia	10 (2.6)	16 (5.7)	0.039	26 (3.9)
Hepatitis	9 (2.3)	12 (4.2)	0.151	21 (3.1)
Hypertension	48 (12.2)	43 (15.2)	0.268	91 (13.5)
Malignancies	19 (4.8)	8 (2.8)	0.186	27 (4.0)
Osteoporosis	13 (3.3)	8 (2.8)	0.718	21 (3.1)
Thyroid disorders	18 (4.6)	22 (7.8)	0.084	40 (5.9)
Extraintestinal manifestations, *n* (%)	80 (20.4)	38 (13.4)	**0.024**	118 (17.5)
Arthropathies	48 (12.2)	25 (8.8)	0.159	73 (10.8)
Erythema nodosum	11 (2.8)	5 (1.8)	0.381	16 (2.4)
Psoriasis	18 (4.6)	7 (2.5)	0.150	25 (3.7)
Pyoderma gangrenosum	3 (0.8)	2 (0.7)	0.930	5 (0.7)
Uveitis	14 (3.6)	-		14 (2.1)
Concurrent non-biologics, median (Q1–Q3)	1 (0–1)	1 (0–1)	0.658	1 (0–1)
Concurrent non-biologics, *n* (%)				
Mesalazine	114 (29.1)	207 (73.1)	**<0.001**	321 (47.6)
Azathioprine	9 (2.3)	8 (2.8)	0.664	17 (2.5)
Corticosteroids	9 (2.3)	6 (2.1)	0.878	15 (2.2)
Mercaptopurine	4 (1.0)	1 (0.4)	0.319	5 (0.7)
Sulfasalazine	8 (2.0)	5 (1.8)	0.798	13 (1.9)
Methotrexate	2 (0.5)	5 (1.8)	0.112	7 (1.0)
Switched, *n* (%)	127 (32.4)	96 (33.9)	0.740	223 (33.0)

* *p* values were calculated comparing patients with CD versus patients with UC (Pearson’s chi-square test or Mann Whitney U test). Bold indicates the statistically significant *p* values. Abbreviations: BMI = Body Mass Index; CD = Crohn’s disease; Q1 = first quartile; Q3 = third quartile; UC = ulcerative colitis.

**Table 2 biomedicines-10-03280-t002:** Basal characteristics of patients affected by IBD for each biologic.

Characteristic	IFX (*n* = 225)	ADA (*n* = 269)	GOL (*n* = 17)	VED (*n* = 145)	UST (*n* = 19)
Sex, *n* (%)					
Females	84 (37.3)	115 (42.8)	10 (58.8)	60 (41.4)	11 (57.9)
Males	141 (62.7)	154 (57.2)	7 (41.2)	85 (58.6)	8 (42.1)
Median age (Q1–Q3)	38 (27–51)	41 (28–55)	49 (39–59)	62 (44.5–71.5)	38 (27–59)
Median age at diagnosis (Q1–Q3)	29 (20–40)	27 (20–40)	34 (24–37.5)	44 (25.5–63.0)	24 (15–37)
Naïve, *n* (%)	209 (92.9)	210 (78.1)	-	7 (4.8)	3 (15.8)
Diagnosis, *n* (%)					
Crohn’s disease	73 (32.4)	249 (92.6)	-	52 (35.9)	17 (89.5)
Ulcerative colitis	152 (67.6)	20 (7.4)	17 (100)	93 (64.1)	2 (10.5)
Disease duration, median (Q1–Q3)	6 (2–13)	8 (3–15)	14 (5–24.5)	10 (4–19.5)	11 (7–17)
Surgery, *n* (%)	21 (9.3)	101 (37.5)	1 (5.9)	21 (14.5)	10 (52.6)
Smoking, *n* (%)					
Smoker	33 (14.7)	79 (29.4)	3 (17.6)	26 (17.9)	4 (21.1)
Ex-smoker	31 (13.8)	35 (13.0)	4 (23.5)	35 (24.1)	1 (5.3)
Non-smoker	161 (71.6)	155 (57.6)	10 (58.8)	84 (57.9)	14 (73.7)
BMI, median (Q1–Q3)	24 (21.6–26.6)	23.5 (20.9–26.4)	27.3 (13.7–31.6)	23.9 (22.7–31.8)	23.1 (20.9–26.1)
Comorbidities, median (Q1–Q3)	0 (0–1)	0 (0–1)	0 (0–1.0)	1 (0–2)	0 (0–1)
Comorbidities, *n* (%)					
Anxiety–depressive disorders	6 (2.7)	14 (5.2)	1 (5.9)	7 (4.8)	-
Cardiovascular disease	18 (8.0)	15 (5.6)	-	23 (15.9)	-
Chronic kidney disease	1 (0.5)	3 (1.1)	-	10 (6.9)	-
Chronic pulmonary disease	5 (2.2)	7 (2.6)	-	3 (2.1)	-
Diabetes	7 (3.1)	5 (1.9)	3 (17.6)	16 (11.0)	1 (5.3)
Dyslipidemia	9 (4.0)	6 (2.2)	-	11 (7.4)	-
Hepatitis	6 (2.8)	5 (1.9)	1 (5.9)	9 (6.2)	-
Hypertension	26 (11.6)	23 (8.6)	-	42 (29.0)	-
Malignancies	-	7 (2.6)	-	20 (13.8)	-
Osteoporosis	5 (2.2)	4 (1.5)	-	10 (6.9)	2 (10.5)
Thyroid disorders	17 (7.6)	12 (4.8)	1 (5.9)	9 (6.2)	-
Extraintestinal manifestations, *n* (%)	38 (16.9)	58 (21.6)	3 (17.6)	13 (9.0)	6 (31.6)
Arthropathies	22 (10.1)	38 (14.1)	2 (11.8)	6 (4.1)	4 (21.1)
Erythema nodosum	7 (3.2)	5 (1.9)	1 (5.9)	3 (2.1)	-
Psoriasis	5 (2.3)	12 (4.5)	-	4 (2.8)	4 (21.1)
Pyoderma gangrenosum	2 (0.9)	3 (1.1)	-	-	-
Uveitis	2 (0.9)	9 (3.3)	1 (5.9)	1 (0.7)	1 (5.3)
Concurrent non-biologics, median (Q1–Q3)	1 (0–1)	0 (0–1)	1 (0–1)	1 (0–1)	0 (0–0)
Concurrent non-biologics, *n* (%)					
Mesalazine	126 (57.8)	92 (34.2)	12 (70.6)	86 (59.3)	2 (10.5)
Azathioprine	7 (3.2)	6 (2.2)	1 (5.9)	2 (1.4)	-
Corticosteroids	8 (3.7)	3 (1.1)	-	3 (2.1)	1 (5.3)
Mercaptopurine	1 (0.5)	3 (1.1)	-	1 (0.7)	-
Sulfasalazine	5 (2.3)	6 (2.2)	1 (5.9)	1 (0.7)	-
Methotrexate	3 (1.4)	3 (1.1)	-	-	-
Switched, *n* (%)	75 (33.3)	93 (34.6)	7 (41.2)	44 (30.4)	-

Abbreviations: ADA = adalimumab; BMI = Body Mass Index; GOL = golimumab; IFX = infliximab; Q1 = first quartile; Q3 = third quartile; VED = vedolizumab; UST = ustekinumab.

**Table 3 biomedicines-10-03280-t003:** Switches and swaps between biologics during the study period.

**Switch from**	**Switch to**						
	IFX		ADA			GOL	
IFX			2 (9)			6 (9)	
ADA	19 (5)					2	
GOL	6		-				
**Swap from**	**Swap to**						
	IFX	ADA		VED	GOL		UST
IFX				29 (15)			3 (2)
ADA				4 (4)			44 (15)
VED	15 (3)	4			5		13 (4)
GOL				1			-

Numbers for switch or swap related to failures; in brackets switch and swap related to ADRs. The cells with grey background indicate a practically impossible switch/swap from/to the same biologic drug. Abbreviations: ADA = adalimumab; GOL = golimumab; IFX = infliximab; VED = vedolizumab; UST = ustekinumab.

**Table 4 biomedicines-10-03280-t004:** Total years of treatment and number of failures and adverse drug reactions for each biologic.

Drug (n. Treatments)	Total Years of Treatment	Number of Total Failures (PF + SF)	Failure/10 yrs	Total Number of ADRs	ADRs/10 yrs
IFX (277)	710.8	54	0.8	95	1.3
ADA (292)	890	80	0.9	84	0.9
GOL (50)	61.3	22	3.6	7	1.1
VED (215)	336	60	1.8	40	1.2
UST (118)	90.3	11	1.2	23	2.5

Abbreviations: ADA = adalimumab; ADRs = adverse drug reactions; GOL = golimumab; IFX = infliximab; PF = primary failure; SF = secondary failure; VED = vedolizumab; UST = ustekinumab; yrs = years.

**Table 5 biomedicines-10-03280-t005:** Type of adverse drug reactions observed during the follow-up period for each biologic.

Type of ADR, *n* (Incidence per 10 Years of Treatment)	IFX	*p* Value *	ADA	*p* Value *	GOL	*p* Value *	VED	*p* Value *	UST	*p* Value *
SADRs ^#^	17 (0.2)	0.258	22 (0.3)	0.666	4 (0.7)	0.926	19 (0.6)	0.461	9 (1.0)	0.989
Infections	22 (0.3)	0.958	26 (0.3)	0.431	4 (0.7)	0.972	13 (0.4)	0.259	10 (1.1)	0.794
Pneumonia	-	-	4 (0.04)	0.127	-	-	2 (0.1)	0.703	1 (0.1)	-
Infusion/injection-related reaction	36 (0.5)	**<0.001**	3 (0.03)	**0.001**	-	-	1 (0.03)	-	2 (0.2)	0.125
Skin reaction	10 (0.1)	0.703	24 (0.3)	**<0.001**	1 (0.2)	-	-	-	3 (0.3)	0.391
Joint disease	7 (0.1)	**0.025**	3 (0.03)	0.670	-	-	2 (0.1)	0.623	-	-
Perianal disease	4 (0.1)	0.592	3 (0.03)	0.807	-	-	3 (0.1)	0.707	1 (0.1)	-
Malignancies	2 (0.03)	0.220	3 (0.03)	0.451	1 (0.2)	-	8 (0.2)	**0.002**	-	-
Blood disorders	2 (0.03)	0.817	3 (0.03)	0.302	-	-	1 (0.03)	-	-	-
Others	11 (0.2)	0.607	15 (0.2)	0.537	1 (0.2)	-	10 (0.3)	0.911	6 (0.7)	0.749

* Patients without ADRs versus patients with ADRs (Pearson’s chi-squared test). Bold indicates the statistically significant *p* values. ^#^ SADRs include hospitalization, life-threatening events, surgery, malignancies and death. Abbreviations: ADA = adalimumab; ADR = adverse drug reaction; GOL = golimumab; IFX = infliximab; VED = vedolizumab; UST = ustekinumab.

## Data Availability

The dataset generated for this study will not be made publicly available.

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
