# Peer review of "Effectiveness and Safety Profiles of Biological Therapies in Inflammatory Bowel Disease: Real Life Data from an Active Pharmacovigilance Project"

_biomedicines, 2022, doi:10.3390/biomedicines10123280_

Round 1

Reviewer 1 Report

The paper describes pharmacovigilance data of patients suffering from inflammatory bowel disease with a particular emphasis on adverse effects. Overall, the paper is well written with good data presentation. Few aspects need to be corrected prior to considering the paper for publication.

1.     The abstract needs modification to highlight the rationale of the study. The research void should be clearly stated.

2.     In methods, the criteria for defining adverse effect/reaction should be clearly mentioned.

3.     The organization of table 3 needs to be improved to follow the data easily

4.     L159-163 and table 4: Indicate how this data is important and linked with the objectives of this study

5.     Regarding the persistent patients, it would be better to have a comment on the choice of drug preferred by the patient/physician that render less switching or stoppage of that drug. Is it related to mainly adverse effects or it is also based on cost effectiveness/availability of the drug? 

Author Response

Please see the attachment with our point-by-point response.

Reviewer 2 Report

I have read with interest this prospective study with active PV monitoring from Italy. The authors had very ambitious goal to analyze safety profile of most commonly utilized biologic agents for treatment of CD and UC;

I think the study is well planned and methodologically executed. It encompasses 675 patients of which they registered 249 SAEs.

I have the following comments and questions:

1.       Line 55-56- Vasculitis have been described as SAE of biologics and this should be included ( please see: https://www.ncbi.nlm.nih.gov/pmc/articles/PMC8836768/)

2.       In Methodology, lines 99: what was the rationale to describe only SAE and not important medical events too?

3.       Line 135- there is a duplicate words, please delete “compared to UC”

4.       Table 1- are these all or the most common extraintestinal manifestations? Have you had any case of vasculitis?

5.       Could you describe from your analysis if patients with EIMs had more SAEs than patients without EIMs of IBD?

6.       Line 150: 223 patients ( 33%) switched therapy- was it due to side effects or lack of medication efficacy?

7.       Section 3.3- in description of adverse effects: infections and malignancy- please discuss if these SAEs are considered by the Sponsor (these medications are in post marketing phase) related or unrelated to medications. If they are unrelated then I think discussion about these should be deleted; If they are related – please expand discussion about mechanisms

8.       Have you noticed any difference in age in patients with SAEs and those without? As you might be aware ageing microbiome might be responsible not only for development of IBD but also for the response ( or lack of response) to therapy and development of ADRs

9.       Line 230- 246: for my liking there is too much discussion about these 2 patients with fatal outcome. I would shorten this. It is too long. Also Sponsor’s assessment regarding causality should be reported

10.   Line 261-285- Please consolidate discussion- it is too long and I feel that too much emphasis is placed on persistence vs non persistence rates and too little to actual description of ADRs that occurred;

11.   Line 331- another hypothesis is “leaky gut syndrome” and it seems logical that bacterial translocation can play the role in development of infection. However, this would be intrinsically related to the IBD itself not medication used to treat it

12.   Cancer development- more discussion should be done regarding the risk for cancer in patients with IBD and how we can differentiate the risk from disease itself vs medication used to treat it.

Author Response

(The authors gave the same response as above.)

Round 2

Reviewer 2 Report

The authors have successfully improved the manuscript. I do not have any further comments